# Implementing Climate Change and Associated Future Timber Price Trends in a Decision Support System Designed for Irish Forest Management and Applied to Ireland's Western Peatland Forests

**Anders Lundholm \*, Edwin Corrigan and Maarten Nieuwenhuis**

UCD Forestry, School of Agriculture and Food Science, University College Dublin, Belfield, Dublin 4, Ireland; edwin.corrigan@ucd.ie (E.C.); maarten.nieuwenhuis@ucd.ie (M.N.)

\* Correspondence: anders.lundholm@ucdconnect.ie

**Abstract:** *Research Highlights:* Predicting impacts on forest management of Climate Change (CC) and dynamic timber prices by incorporating these external factors in a Forest Management Decision Support System (FMDSS). *Background and Objectives:* Forest managers must comply with Sustainable Forest Management (SFM) practices, including considering the long-term impacts that CC and the bioeconomy may have on their forests and their management. The aims of this study are: (1) incorporate the effects of CC and Dynamic Prices (DP) in a FMDSS that was developed for Ireland's peatland forests, (2) analyse the impact of global climate and market scenarios on forest management and forest composition at the landscape level. *Materials and Methods:* Remsoft Woodstock is a strategic planning decision support system that is widely used for forest management around the world. A linear programming model was developed for Ireland's Western Peatland forests while using Woodstock. Data from Climadapt, which is an expert-based decision support system that was developed in Ireland, were used to include CC effects on forest productivity and species suitability. Dynamic market prices were also included to reflect the changing demands for wood fibre as part of the European Union (EU) and global effort to mitigate CC. *Results:* DP will likely have more impact on harvest patterns, volumes, and net present value than CC. Higher assortment prices, especially for pulpwood, stimulate the harvesting of forests on marginal sites and off-set some of the negative CC growth impacts on forest profitability. *Conclusions:* Incorporating CC and bioeconomy prices in a forest decision support system is feasible and recommendable. Foresters should incorporate the expected global changes in their long-term management planning to mitigate the negative effects that un-informed management decisions can have on the sustainability of their forests.

**Keywords:** linear programming; optimization; bioeconomy; Woodstock; Remsoft spatial planning software

## 1. Introduction

Industrialism and a growing agrarian population had reduced Ireland's forest cover to 1.5% in 1908 [1]. The necessity of a domestic timber supply caused the Irish state to initiate an afforestation programme, which largely focused on purchasing agriculturally marginal land for public afforesting [2–4]. Since the late 1990s, afforestation has been primarily on private land and the forest cover in 2017 reached 11%, or 770,020 ha [5]. Inexpensive and agriculturally marginal land often meant mountainous blanket peat. Ploughing for draining and the application of phosphatic rock fertiliser for mineral nutrients enabled the establishment of hardy, fast-growing conifers from Western North America on the wet and nutrient poor blanket peat [6]. In 2012, 35.6% of the public forest and 30.2% of

the private grant-aided forest was situated on blanket peat [7]. Sitka spruce (*Picea Sitchensis* (Bong.) Carr.) has a high Yield Class (YC is the maximum mean annual increment of cumulative timber volume production for a species on a site, as given in $m^3$ $ha^{-1}$ $year^{-1}$. Mean annual increment starts at zero and it increases as the forest stand grows older; after it reaches maximum mean annual increment, i.e., the YC, mean annual increment declines. Irish plantation conifers reach the maximum mean annual increment between 50 and 60 years of age, depending on the species and site conditions) on a wide range of site types [8], and it is still favoured for this reason in Irish forestry, covering 51.1% of the Irish forest estate [5]. Lodgepole pine (*Pinus contorta* Douglas) was chosen for the least productive sites and it occupies 9.6% of the Irish forest estate [5].

Forests are susceptible to climate change (CC); due to the long lifespans of trees they must be able to live and grow in both the current and future climates [9]. The influence of CC on Ireland, under a high emission scenario (3.7 °C average global temperature increase by 2100 compared to pre-industrial levels) will likely result in a warming of the weather, increasing both the summer and winter average temperature by 2 and 2–3 °C, respectively [10]. The precipitation patterns are likely to change, with up to 20% reduction in summer precipitation and 14% increase in winter precipitation. Heavy rain events are expected to increase by 20% and storms and floods will be more common. Increased $CO_2$ levels and temperatures would generally be expected to result in higher biomass productivity, and thus forest growth, for most of the country [9,11]. However, when considering future precipitation patterns, soil types, and species response, it is unlikely that all of the species in Ireland will experience increased growth rates in the future. Forests in eastern Ireland will likely suffer more droughts, as the area is expected to experience the highest temperature increase and largest reduction in precipitation [11]. The Irish growing season extends well into the winter months, averaging 250–300 days over the island [12], and most of the forests are located on wet soils [5]. Sitka spruce develops shallow root plates when growing on wet soils, which causes decreased wind stability and limits the depth at which roots can uptake nutrients [13]. The Irish software Climadapt [14] was developed to predict the future species suitability and YC under different CC scenarios. The predictions are based on Ecological Site Classifications (ESCs) [15] and the International Panel for Climate Change (IPCC) climate change predictions. The ESCs are based on the edaphic condition of soil nutrient regime and soil moisture regime, and the climatic factors of accumulated temperature, moisture deficit, a detailed aspect method of scoring (for wind and exposure), and continentality. The main ESCs that will change and impact Irish forestry—according to Climadapt—are accumulated temperature and soil moisture regime (summer and winter) [14].

Sustainable forestry was developed in response to local deforestation resulting from the mining industry in the early 18th-century, which caused concerns regarding charcoal supply [16]. The sustainability concept has been expanded to include economic, ecological, and social values in Sustainable Forest Management (SFM), as defined by the United Nations Conference on Environment and Development in Rio de Janeiro in 1992 [17,18]. Modern forest managers in Ireland must comply with SFM practices, and in doing so need to consider the long-term impacts that climate change might have on forests, regarding productivity, species suitability, and forest resilience to extreme weather, pests, and diseases. Simulation and optimization algorithms were first used in forest management in the 1960s [19], and Forest Management Decision Support Systems (FMDSSs) became popular for transferring scientific knowledge to practical forest management in the 1980s [20,21]. The FMDSSs were initially developed to deal with sustainable timber yield and optimal harvest scheduling [19,22]. Timber production or Net Present Value (NPV) often remain the main focus of these systems [21,23]. However, forest companies must adhere to the SFM concept, environmental regulations, and forest certification. Thus, FMDSSs have been developed to analyse the long-term CC effects on forests, including the impacts of disease, pest and windthrow damage, as well as impacts on biodiversity, carbon sequestration, water quality, and the long-term changes in forest structure [19,22,24,25].

There are multiple ways in which forest planning and FMDSSs can be categorized. The length of the planning period, the detail of the plan, and the focus of the plan are often divided into three

hierarchical levels; strategic (40–100 years), tactical (3–10 years), and operational (weekly—A year) [23]. Strategic planning, as utilised in this study, often focuses on the sustainable long-term production of timber, given the bio-physical and policy constraints. The spatial location and grouping of potential harvestable areas from where to source wood is part of tactical planning; what weeks stands should be harvested and which machines to use is addressed at the operational planning level. FMDSSs primarily operate based on either simulation or optimization. The simulation uses a pre-defined sequence of actions to manage forest stands and reports the outcome over the planning period. Optimization can evaluate the outcome from several available actions, and given an objective and constraints, reports the optimal sequence of actions to implement [26]. While, with simulation, you decide which management prescription to implement, and the FMDSS report on their long-term results; in optimization, you decide the desired long-term results, and the FMDSS reports which management prescriptions should be implemented to achieve those results. Remsoft spatial planning software (Remsoft, Fredericton, Canada) is a suite of DSSs that are globally used to manage over 202 million ha of forests in 15 countries, mainly in the Americas, but also in Ireland. Woodstock [27] is the strategic planner and flagship of Remsoft spatial planning software. Woodstock can utilise both simulation and optimization, and it only provides the user with a modelling structure. Growth and yield tables, price lists, forest management actions, desired forest outputs, and management objectives are user defined, allowing for much customizability in what the developed FMDSS contains and how it operates.

Tree growth and yield models have been developed in Eastern Canada that can project tree growth under both current and future climates, i.e., dynamic growing conditions [28]. This was achieved by combining empiric growth data and growth projections from an ecological process-based model that uses a range of CC scenarios. These models reduce the uncertainty in future management decisions regarding when and where to harvest. Another Canadian study found that long-term changes in forest growth, which are due to CC, have more of a significant impact on future boreal forest structure than both harvesting and natural disturbances [29]. The growth effects were mainly due to the climate becoming too warm for the cold-adapted boreal tree species. These expected changes in growth differ from the Irish estimations. However, it indicated that forecasting forest growth under changing climate is possible and that the use of regionally specific CC data is critical when forecasting the future forest conditions. A study utilised Climadapt to evaluate the growth potential of Sitka spruce under high emission CC scenarios in Ireland and found that a growth reduction of about 25% can be expected nationally by 2080 [30]. The reductions were mainly water related, with the southern areas suffering up to 37% growth reduction due to moisture deficits, and western areas only suffering a 14% growth reduction due to prolonged waterlogging in the autumn and winter. Another Irish study implemented Climadapt's A2 CC scenario future YC predictions in Woodstock to evaluate long-term CC impacts on forestry in western Ireland [31]. Their model was run three times: without YC change, YC changed in year 2050 (future climate based on the period 2020–2050), and YC changed in year 2080 (future climate based on the period 2050–2080). The NPV of forestry was reduced when changing the YC, but the NPV was reduced more in the 2050 YC prediction than in the 2080 YC prediction. The 2080 YC change had more of a negative effect on the productivity, but the earlier implementation of CC's negative impact on the growth of commercially valuable species had a stronger negative impact on the NPV.

Climate change, population development, economic growth, and policy are interlinked factors that determine the future for this study. Adverse changes in climate could have a devastating economic impact to natural resource production systems and population development [32]. However, economic growth in renewable energy sectors and higher utilisation of renewable resources, rather than fossil fuel-based, could mitigate CC [33]. Properly formulated policy (at the national, European Union (EU), and global level) could play an important role in reducing emissions, and thus mitigating CC. The forestry sector could have an important role in CC mitigation and the bio-economy by supplying biomass for energy, fuel, packaging and construction [34]. The behaviour of forest owners to intensify wood extraction will largely depend on changing timber prices and policy [35].

The aim of this study was to: (1) incorporate, on an annual basis, climate change and dynamic timber prices in a FMDSS that was developed for Ireland; and, (2) analyse the impact that global change scenarios (representing different levels of CC and mitigation efforts) will have on forest management approaches and forest landscape composition.

## 2. Materials and Methods

### 2.1. Case Study Area Description—Barony of Moycullen

The Barony of Moycullen in County Galway, Ireland, was chosen as the Case Study Area (CSA) (Figure 1). The CSA contains the Cloosh Valley forest and Derrada forest, with parts of Cloosh and parts of Derrada forming the largest continuous forest area in Ireland, at almost 4600 ha. Sitka spruce and lodgepole pine in the CSA were mainly planted in the 1970s and 1980s. The area has high recreational pressure, both from tourists and the residents of Galway city, and it contains one of Ireland's priority eight Freshwater Pearl Mussel (*Margaritifera margaritifera* L.) catchments [36]. These priority catchments hold 80% of Ireland's freshwater pearl mussel numbers and they are important for the long-term survival of the species. The Cloosh and Derrada forests are surrounded by Natura2000 designated areas to protect bog habitat and the freshwater pearl mussel catchment. These multiple-uses and their interactions make the forested landscape an interesting testing ground for the analysis of the long-term sustainability impacts resulting from CC and the associated changes in timber prices. The CSA is representative for forestry in western Ireland and the nation's peatland forests. The CSA was chosen as a research project that investigates new forest management methods that could provide a wider range of ecosystem services under different future scenarios. Blanket peat soil is extremely poor in mineral nutrients and this, coupled with the CSA's proximity to the Atlantic Ocean, causes windthrow to be a serious problem in forest management. Table 1 provides the descriptive CSA statistics.

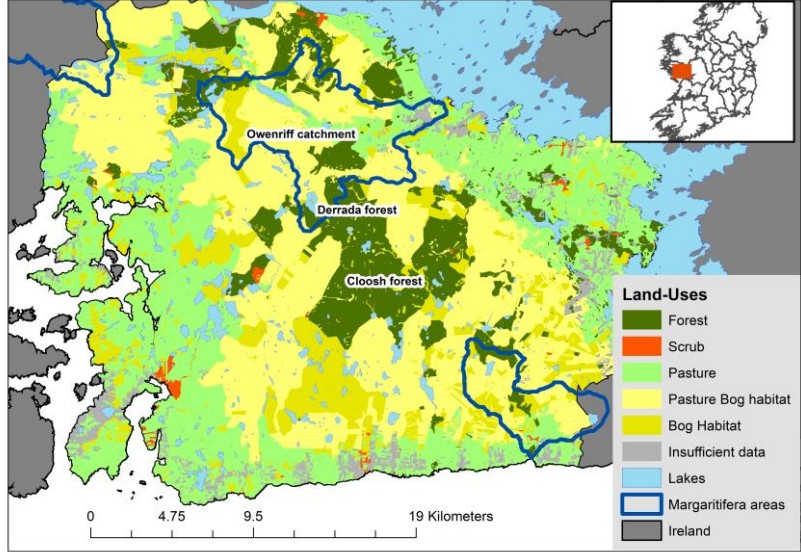

**Figure 1.** The Barony of Moycullen Case Study Area (CSA) in County Galway, Ireland, delineated by its land-uses. Pasture Bog Habitat is bog utilised for commonage pastures. Margaritifera areas refer to catchments with freshwater pearl mussels. The Cloosh Forest is the forest area at the centre of the CSA, Derrada is located north of Cloosh. The Owenriff catchment contains much of the Derrada forest and it is located just north of Cloosh.

**Table 1.** Descriptive statistics for the case study area relating to climate (1981–2010), land-use, and forest data. Climate data from [37], land-use, forest, and soil data from the GIS sources listed in Table 2.

| | **Barony of Moycullen** | |
|---|:---:|:---:|
| Temperature July mean range (°C) | 12–19 | |
| Temperature January mean range (°C) | 3–8 | |
| Annual rainfall range (mm) | 1600–2000 | |
| | **ha** | **%** |
| Forest area | 10,230.0 | 13.2 |
| Enclosed pasture | 4882.5 | 6.3 |
| Blanket bog | 8447.5 | 10.9 |
| Bog land/commonage pasture | 45,260.0 | 58.4 |
| Scrub | 465.0 | 0.6 |
| Other and urban | 8215.0 | 10.6 |
| Total area | 77,500.0 | 100 |
| **Ownership of Forest (%)** | | |
| Public (Coillte) | 81.1 | |
| Private | 18.9 | |
| **Species Cover by Area (%)** | | |
| Sitka spruce | 41.0 | |
| Lodgepole pine | 29.4 | |
| Other conifers | 4.1 | |
| Broadleaves | 6.0 | |
| Open area in forest stands | 19.5 | |
| **Average Standing Volume (m³ ha⁻¹)** | | |
| Forest | 165 | |
| **Productivity (YC) by Stocked Forest Area (%)** | | |
| ≤10 | 33.6 | |
| 12–14 | 44.0 | |
| 16–18 | 20.0 | |
| ≥20 | 2.4 | |
| **Age Class Distribution by Forest Area (%)** | | |
| ≤10 | 8.8 | |
| 11–20 | 15.5 | |
| 21–30 | 29.4 | |
| 31–40 | 35.1 | |
| 41–50 | 6.8 | |
| ≥51 | 4.4 | |
| **Soils** | **Forest Area (%)** | **CSA (%)** |
| Lithosols/Peaty podzols | 15.3 | 18.7 |
| Blanket peats | 82.0 | 71.7 |
| Gleys | 0.2 | 1.3 |
| Brown earths/podzolics | 1.7 | 4.2 |
| Cutaway peat | 0.8 | 4.1 |
| **Elevation** | **Area (%)** | **CSA (%)** |
| 0–100 | 50.4 | 75.9 |
| 101–200 | 40.7 | 20.1 |
| ≥201 | 8.9 | 3.9 |

The table header for Average Standing Volume uses units $m^3\ ha^{-1}$.

**Table 2.** GIS data sources used to create the land-use layer that was imported into Woodstock.

| GIS Data | Source |
| --- | --- |
| Coillte forest inventory—updated May 2016 | Coillte |
| County council roads | Coillte |
| Wind zones | Coillte |
| Private forest—Forests2015—updated December 2016 | Forest Service |
| Single farm payments | Department of Agriculture, Food and the Marine |
| Corine land classification | Environmental Protection Agency (EPA) |
| Teagasc soil survey—Irish forest soils | EPA |
| River, waterbodies and catchments | EPA |
| Native woodland sites | National Parks and Wildlife Services (NPWS) |
| Ancient and long-established woodland | NPWS |
| Natura 2000 sites | NPWS |
| Margaritifera sensitive areas | NPWS (special request) |
| Digital elevation model | UCD Urban Institute |

*2.2. Decision Support System and Model Setup*

The forecasting model was built in Woodstock (64-bit), version 2017.11.0 of the Remsoft Spatial Planning System. The FMDSS was run in a Windows 10 Professional 64-bit operating system with an Intel® Core™ i7-3930K CPU @ 3.20 GHz (six cores with two threads per core) PC with 32 GB of RAM.

Woodstock can utilise linear-programming, is suitable for strategic forest planning, and has broad user customizability. Strategic planning in Woodstock allows for the user to determine the spatial constraints before the optimization, e.g., harvest scheduling is not restricted by the spatial adjacency of stands, but areas that are not eligible for harvesting can be selected beforehand. Woodstock was setup to use an ESRI (Environmental Systems Research Institute, Redlands, California, United States) shapefile for forest inventory information. The user sets up themes, which describe forest stand characteristics and they are crucial for Woodstock to link stands to yield tables and determine forest management prescription eligibility (Figure 2). Once the model is complete, Woodstock is used to generate a linear programming matrix, in mathematical programming system format, with all possible forest management actions. The matrix, including the objective function, forms the basis of the linear programming model that is solved while using mathematical programming solver software (in this study, MOSEK (Copenhagen, Denmark)). The basic Woodstock model that was built for forestry requires a forest inventory, forest management prescriptions, yield tables, and an objective function to build a matrix and generate outputs of the future forest landscape condition. The remainder of this Material and Methods section focuses on combining the GIS data to build a forest inventory shapefile that is appropriate for integrating into Woodstock for strategic level planning, the growth and yield tables that were used, the eligibility for forest management prescriptions, including prescription costs and timber prices, and the implementation of the global scenarios.

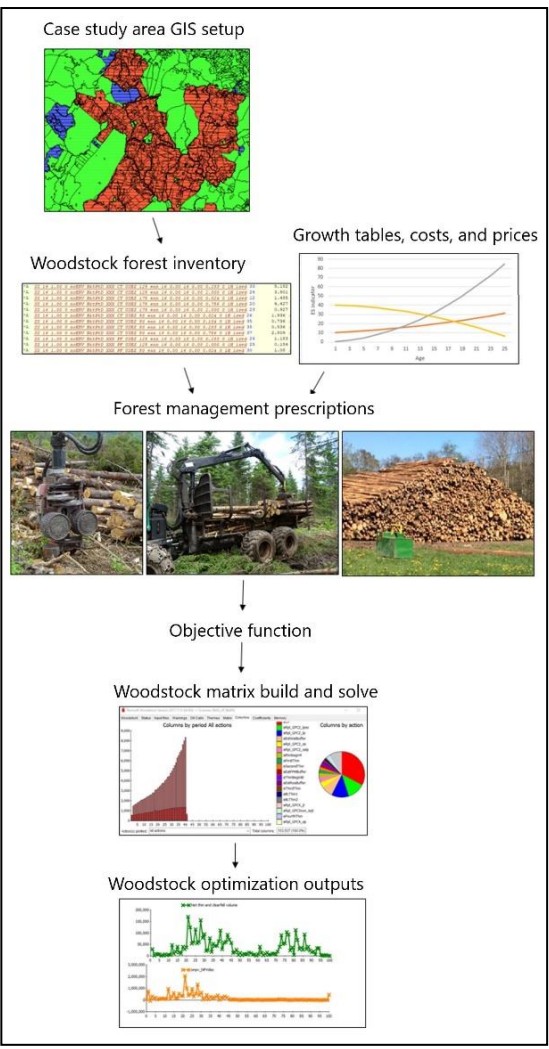

**Figure 2.** Components required to build a Woodstock forest management decision support system model and generate and solve the matrix to generate outputs for reporting.

### 2.3. Coillte Collaboration and UCD Woodstock Model

Coillte (the Irish state forestry board, who manage Ireland's publicly-owned forests) provided their Woodstock model to University College Dublin in February 2012 that was to be used in a research project [24]. Since May 2016, this Woodstock model [38] has undergone a material development phase for use in this study. The changes to the model include updated costs, revenues, growth and yield tables, policy rules, and the implementation of global climate and price change scenarios.

### 2.4. GIS Setup

ArcGIS 10.4 was used to produce the ESRI shapefile containing spatial location of forest stands and the linked database with site information and forest inventory data. Merging vector layers from multiple sources created a shapefile with information on land-uses, i.e., Coillte forests, private forests, agricultural, scrub, and peatland created the imported shapefile (Table 2). Attributes necessary for the running of the Woodstock model were assigned to polygons using ArcGIS 10.4, these were: species, site productivity, species percentage of stand, soil type, elevation, policy, environmental and native woodland designations, eligibility for site preparation, thinning status, polygon status, hydrological catchment, and stand age. Data modification was also done in Woodstock, e.g., aggregating elevation above sea level to silviculturally relevant ranges and the grouping and classification of species for which no yield tables were available as an alternative species (see Table 3).

**Table 3.** Tree species used in the model, the origin of their growth and yield tables, and the minimum and maximum yield class (m$^3$ ha$^{-1}$ year$^{-1}$) for each species based on available growth and yield tables.

| Species (Groups) | Latin Name | Growth Table | Min YC | Max YC |
|---|---|---|---|---|
| Douglas fir | *Pseudotsuga menziiesi* (Mirb.) Franco | GROWFOR | 4 | 24 |
| Norway spruce | *Picea abies* (L.) H. Karst. | GROWFOR | 4 | 22 |
| Scots pine | *Pinus sylvestris* L. | GROWFOR | 4 | 14 |
| Sitka spruce | *Picea Sitchensis* (Bong.) Carr | GROWFOR | 4 | 24 |
| Noble fir/Other conifers | *Abies nobilis* (Rehder) | GROWFOR | 4 | 22 |
| Lodgepole pine | *Pinus contorta* (Douglas) | GROWFOR | 4 | 16 |
| Larch | *Larix* | British Forestry Commission | 4 | 14 |
| Beech | *Fagus sylvatica* L. | British Forestry Commission | 4 | 10 |
| Oak | *Quercus* | British Forestry Commission | 4 | 8 |
| Downy Birch/Other broadleaves | *Betula pubescens* Ehrh. | British Forestry Commission | 4 | 14 |

*2.5. Growth and Yield Tables*

The Woodstock model accommodates up to three species per stand; the percentage and YC of each species are represented as themes in the model. Additionally, the model assumes 15% open space in each stand, which represents unproductive area; this is an industry standard metric. GROWFOR was used to create the growth and yield tables for most conifer species [39], where data were available. Larches and broadleaf species utilised British Forestry Commission tables [40], since insufficient Irish forest data were available to produce GROWFOR yield tables for these species. Table 3 presents a full list of tree species used in the model and their YC ranges. Sitka spruce YC is used in Irish forestry to compare the growth potential of Sitka spruce to that of other species on the same site. The corresponding YC for other species, which is based on the Sitka spruce YC, used in the model can be found in Table 4. Sitka spruce YC were used in the Woodstock model to maintain the sites' field estimated productivity when changing species during reforestation, rather than estimating the YC using a generic equation.

**Table 4.** Lookup table to determine the Yield Class (YC) (m$^3$ ha$^{-1}$ year$^{-1}$) of species based on the Sitka spruce YC [41].

| | Sitka Spruce YC | | | | | | | | | | | | |
|---|---|---|---|---|---|---|---|---|---|---|---|---|---|
| Corresponding species YC | 6 | 8 | 10 | 12 | 14 | 16 | 18 | 20 | 22 | 24 | 26 | 28 | 30 |
| Douglas fir | 6 | 8 | 10 | 10 | 12 | 14 | 16 | 16 | 18 | 18 | 20 | 22 | 24 |
| Japanese larch | 4 | 6 | 6 | 8 | 8 | 10 | 10 | 10 | 12 | 12 | 14 | 14 | 14 |
| Lodgepole pine | 6 | 8 | 8 | 10 | 10 | 12 | 12 | 12 | 12 | 12 | 12 | 14 | 14 |
| Noble fir | 6 | 8 | 10 | 12 | 12 | 14 | 16 | 16 | 18 | 20 | 20 | 22 | 22 |
| Norway spruce | 6 | 8 | 10 | 10 | 12 | 14 | 16 | 18 | 20 | 20 | 22 | 22 | 22 |
| Scots pine | 4 | 6 | 8 | 8 | 10 | 10 | 12 | 12 | 12 | 14 | 14 | 14 | 14 |
| Alder (*Alnus glutinosa* (L.) Gaertn.) | 4 | 4 | 4 | 6 | 6 | 6 | 8 | 8 | 8 | 10 | 10 | 12 | 12 |
| Ash (*Fraxinus excelsior* L.) | 4 | 4 | 4 | 6 | 6 | 8 | 8 | 8 | 10 | 10 | 12 | 12 | 12 |
| Beech | 4 | 4 | 4 | 4 | 4 | 6 | 6 | 6 | 8 | 8 | 10 | 10 | 10 |
| Birch | 4 | 4 | 4 | 4 | 6 | 6 | 6 | 8 | 8 | 8 | 10 | 10 | 12 |
| Oak | 4 | 4 | 4 | 4 | 4 | 6 | 6 | 6 | 8 | 8 | 8 | 8 | 8 |
| Sycamore (*Acer pseudoplatanus* L.) | 4 | 4 | 4 | 6 | 6 | 8 | 8 | 8 | 10 | 10 | 12 | 12 | 12 |

*2.6. Forest Management Prescriptions*

Woodstock evaluates the outcome of executing every possible management prescription in all eligible years, i.e., the matrix build. Subsequently, it chooses the prescription schedule that fulfils the objective, i.e., the solve. Management prescriptions are defined based on eligibility, which is based on polygon attributes. The model includes site preparation, reforestation (including buffer establishment), thinning, clearfelling, and retention, and all are defined in Table 5. The reforestation costs were set to €2589 ha$^{-1}$ for conifers and €3281 ha$^{-1}$ for broadleaves. Table A1 presents the cost associated with each reforestation activity. The revenue from the harvested trees was calculated using the millgate price: standing wood value minus costs for felling and extraction, road maintenance and haulage. The standing wood prices were based on a fixed pulpwood price for lodgepole pine (lodgepole pine is only utilised for pulpwood assortment in Ireland [4]), a fixed firewood price for broadleaf volume, and average tree size for all conifers, excluding lodgepole pine (Table A2). The felling and extraction costs were based on the average tree size and they varied between species and harvest operation (clearfelling or thinning) (Table A3). Road maintenance cost was fixed at €1.50 per extracted m$^3$. The haulage costs were based on a Coillte equation that uses transportation distance from the CSA centroid to the respective processing plants to calculate haulage cost per tonne wood. This was converted to haulage cost per m$^3$ using wood density tables [42]. Table A4 presents the haulage costs per species used in the model. Additional costs that are associated with administration and enhanced environmental considerations during harvesting operations were included in the model (Table A5) and they are incurred for: Special Areas of Conservation (EU Habitats directive), Special Protection Areas (EU Birds directive), freshwater pearl mussel catchments, national heritage areas, proposed national heritage areas, peat soils, and buffer zones. Forests on peat soils become unstable after thinning, causing increased windthrow risk, so thinning is not practiced on blanket peat sites in the CSA [43]. Due to freshwater pearl mussel populations in surrounding catchments and areas that are protected by EU bird and habitat directives, an added level of environmental consideration influences forest management in the area. This restricts the aerial fertilisation of forest stands. Manual fertilization is still possible, but it is not performed in the CSA, as it requires more labour than is available [44]. Sitka spruce can be established on drained and fertilised blanket bog sites [6,45], but without fertilization, the species choice on blanket bog is limited to lodgepole pine. Thus, future blanket bog forest management in the model was limited to the planting of lodgepole pine and the clearfelling of conifer plantations. On mineral soils, broadleaf and conifer establishment was an option.

**Table 5.** Forest management prescriptions included in the model and the stands that are eligible.

| Forest Management Prescription | Stand Attributes for Management Prescription Eligibility |
|---|---|
| **Site Preparation** | |
| Mounding/trenching, drain clearing/digging | Applied to all stands at reforestation. Mounding or trenching performed on mineral soils, drain clearing/digging on peat sites. |
| Fencing | Necessary to keep out deer and sheep from plantations. |
| Fertilisation | Although common in Irish forestry, Coillte does not apply fertiliser in the CSA. The sensitivity of Natura2000 designated areas and freshwater pearl mussel catchments mean aerial fertiliser application is not permitted, and there is not enough manpower to fertilise manually. |
| **Forestation** | |
| Reforestation | The Grant Premium Categories [46] were included as a basis for eligible reforestation options. Soil and elevation eligibility were based on Horgan et. al. 2003 [47]. All broadleaves and conifers are eligible on good to medium mineral soils, Sitka spruce and lodgepole were also eligible on poor mineral soils, and lodgepole pine was eligible on peat soils. Broadleaves were eligible up to 120 m elevation (per Coillte recommendation for Western Ireland), all conifers were eligible up to 200 m, Sitka spruce was eligible up to 300 m, and only lodgepole pine was eligible above 300 m. Vegetation control and weevil control were assumed to be applied to all plantations, as was inspection of plantation to ensure successful establishment. |
| Native woodland establishment | The 4 native woodland establishment schemes [48] were included, only eligible on better mineral soils up to 200 m elevation (the 120m limit for Western Ireland being ignored since the purpose is not a commercial forest). |
| Buffer zone | Buffer zones are established adjacent to roads, freshwater pearl mussel watercourses and watercourses to create an unmanaged area protecting sensitive features. Widths vary between 10–25 m, depending on slope, soil type, and protected feature. Buffers are planted with a mix of broadleaves and open space for natural regeneration of native trees. Road buffers have 30% birch, 20% alder, 50% open area, freshwater pearl mussel watercourse buffers have 20% birch and 80% open area, watercourse buffers have 100% open area. In forest stands established before buffer zones were a requirement, the appropriate buffer zone width from watercourse is split from the original forest stand and planted only when a harvest operation is carried out in the main stand. |
| **Harvesting** | |
| Thinning | Conifer stands with YC 14 or higher were eligible for thinning. Thinning was applied on a four-year cycle and up to four times, starting at age 19, 22, and 25. Sitka sprue had additional thinning start ages at 20 and 23. No conifers on blanket peat soil and no lodgepole pine were eligible for thinning [43,49]. Broadleaf thinning for private forests was eligible for up to seven thinnings starting at age 13 on a 10-year interval for Sitka spruce YC 22–30; up to six thinnings starting at age 20 on 15-year interval for Sitka spruce YC 16–20; and up to six thinnings starting at age 28 on 20-year interval for Sitka spruce YC 12–14. Indefinite continuous cover forestry thinning on five-year interval was eligible for all broadleaf stands, starting age 15. Thinning history was assigned to stands in the GIS setup phase. |
| Clearfell | Clearfelling of a stand was possible when the dominant conifer's mean height was between 18 and 26 m, private conifer stands had to be at least 21 years old, as this is when they stop receiving State premia. Private broadleaf stands had to be $\geq$60 years to be eligible for clearfelling, Coillte broadleaves are not eligible for clearfelling due to company policy. |
| Forest retention | No prescription is an option to all stands. They mature without intervention. |

*2.7. Global Scenarios*

There were the three global scenarios that were modelled for this study. They include the effect of climate change on tree growth and dynamic changes in timber assortment prices, for different assumed levels of mitigation efforts. The scenario narratives [50] were provided by the International Institute for Applied Systems Analysis (IIASA), while using the Global Biosphere Management Model (GLOBIOM) [51]. GLOBIOM computes the market equilibrium for agriculture, forestry, and bioenergy, based on land-use competition, population dynamics, global trade, and policies. The model includes the accounting of greenhouse gas emissions and can, as an example, be used to analyse how global development and policy scenarios will affect greenhouse gas emissions in the future. Although GLOBIOM incorporates agricultural adaptation to CC, GLOBIOM did not change forest productivity as a result of CC when producing the dynamic timber assortment prices for this assessment. GLOBIOM provided data on average global temperature increases for each scenario, which was used to find a corresponding CC scenario in Climadapt for changing forest productivity. The global scenarios were based on analyses that combined the European Union policy scenarios [52] and the framework for Representative Concentration Pathways-Shared Socio-economic Pathways (RCP) developed for the International Panel for Climate Change (IPCC) [53]. The climate model that was used was HadGEM2-ES [54,55]. The global scenarios used in this study, including their GLOBIOM climate scenario and descriptions, were:

- BAU—Business As Usual. No CC or dynamic prices (DP) implemented;
- S1—Reference: Temperature increase of 3.7 °C by 2100, compared to pre-industrial values. Climate scenario: RCP8.5. No effort to mitigate CC;
- S2—EU Bioenergy: Temperature increase of 2.5 °C by 2100, compared to pre-industrial values. Climate scenario: RCP4.5. EU effort to mitigate CC through increased bioeconomy; and,
- S3—Global Bioenergy: Temperature increase of 1.5–2.0 °C by 2100, compared to pre-industrial values. Climate scenario: RCP2.6. Global effort to mitigate CC through increased bioeconomy;

2.7.1. Climate Change

Climadapt [14], an Irish CC software, was used to predict the future YC for 11 common Irish forestry species for the year 2080 (Table 6) based on the average climate for the period 2050–2080, using the IPCC A2 scenario [56], which corresponds to RCP8.5. A2, was the only scenario with sufficient information for long-term projection in Climadapt. Based on the relative YC change in the Climadapt prediction, a species-specific area weighted average YC change was calculated for the forested land in the CSA. Rather than assuming all change in the year 2080, the average YC change value was linearly interpolated between 1990, the base year for Climadapt's climate data, and 2080. Due to 2080 being the last year for which YC is predicted in Climadapt, all of the subsequent years were assigned the 2080 productivity change. The Woodstock model start year was 2016, thus a degree of CC had already taken place from the Climadapt start year. CC was implemented in the model by scaling all the volume outputs with the species-specific YC change value. The BAU scenario was run without any CC, the full YC change in the A2 scenario was used for the S1 (Reference) forecast, half the YC change in A2 was used for S2 (EU Bioenergy), while the S3 (Global Bioenergy) scenario experienced the full A2 climate change between 1990–2016, and then experienced no further change.

**Table 6.** Proportional YC for 11 species, for three future scenarios in response to CC. Proportional YC of 100% in 2080 means that the YC is unchanged, a value smaller than 100% means YC decreases, a value higher than 100% means YC increases. The proportional YC change was interpolated between 1990 and 2080.

| Species (Group) | Year | S1—Reference | S2—EU Bioenergy | S3—Global Bioenergy |
|---|---|---|---|---|
| | | **Proportional YC** | | |
| Sitka spruce | 1990 | 100.0 | 100.0 | 100.0 |
| | 2016 | 94.8 | 94.8 | 94.8 |
| | 2080 | 82.1 | 91.1 | 94.8 |
| Lodgepole pine | 1990 | 100.0 | 100.0 | 100.0 |
| | 2016 | 102.0 | 102.0 | 102.0 |
| | 2080 | 106.8 | 103.4 | 102.0 |
| Birch | 1990 | 100.0 | 100.0 | 100.0 |
| | 2016 | 100.7 | 100.7 | 100.7 |
| | 2080 | 102.6 | 101.3 | 100.7 |
| Oak | 1990 | 100.0 | 100.0 | 100.0 |
| | 2016 | 99.0 | 99.0 | 99.0 |
| | 2080 | 96.7 | 98.3 | 99.0 |
| Larch | 1990 | 100.0 | 100.0 | 100.0 |
| | 2016 | 100.0 | 100.0 | 100.0 |
| | 2080 | 100.2 | 100.1 | 100.0 |
| Scots pine | 1990 | 100.0 | 100.0 | 100.0 |
| | 2016 | 114.7 | 114.7 | 114.7 |
| | 2080 | 149.5 | 124.7 | 114.7 |
| Norway spruce, Douglas fir, & Other conifers | 1990 | 100.0 | 100.0 | 100.0 |
| | 2016 | 97.4 | 97.4 | 97.4 |
| | 2080 | 90.9 | 95.4 | 97.4 |
| Beech | 1990 | 100.0 | 100.0 | 100.0 |
| | 2016 | 118.5 | 118.5 | 118.5 |
| | 2080 | 164.2 | 132.1 | 118.5 |
| Sycamore | 1990 | 100.0 | 100.0 | 100.0 |
| | 2016 | 96.6 | 96.6 | 96.6 |
| | 2080 | 88.2 | 94.1 | 96.6 |
| Ash | 1990 | 100.0 | 100.0 | 100.0 |
| | 2016 | 94.9 | 94.9 | 94.9 |
| | 2080 | 82.2 | 91.1 | 94.9 |
| Alder | 1990 | 100.0 | 100.0 | 100.0 |
| | 2016 | 98.2 | 98.2 | 98.2 |
| | 2080 | 93.8 | 96.9 | 98.2 |

2.7.2. Dynamic Assortment Prices

The CC scenarios also involve dynamic assortment prices for pulpwood and sawlog that reflect changing demands for wood fibre associated with the different levels of CC mitigation in the three scenarios. Wood assortment prices increase due to higher demand and the usage of wood fibre to reduce the impact of CC. The DP were produced by IIASA, using GLOBIOM, and they were based on external projections of wood demand in Ireland. The wood demand barely increased for S1 but the increases for S2 and S3 were in line with national projections of future wood harvest [57]. The national wood harvest increases rely on continued afforestation and the maturing of recently afforested private forests; however, there has been very little afforestation in the CSA recently and most of the non-forested sites are not eligible for afforestation. Thus, achieving these increases in harvested wood volumes was not explicitly used as a constraint in the Woodstock model. The prices were delivered as

decennial values for the period 2010 to 2100. The delivered DP differed from those that were outlined by Teagasc [58], so a change factor was calculated and then interpolated linearly between the decennial values (Figure 3). The change factor was then multiplied with the standing volume value for the respective year to implement dynamic assortment prices in the Woodstock model (Table A6). The BAU scenario was run without DP. Generally, both the sawlog and pulpwood prices increased over the planning horizon. The Sawlog prices in S1 increased in the first 30 years and then plateau for the remainder of the planning horizon. Sawlog prices were the highest in S3, and they had a steep price increase towards the end of the planning horizon in S2 and S3. Pulpwood prices increased dramatically by 2030 in all scenarios, and from 2030 to 2060, the S1 and S2 pulpwood prices decreased and then remained static. The S3 pulpwood prices increased each decade throughout the planning horizon.

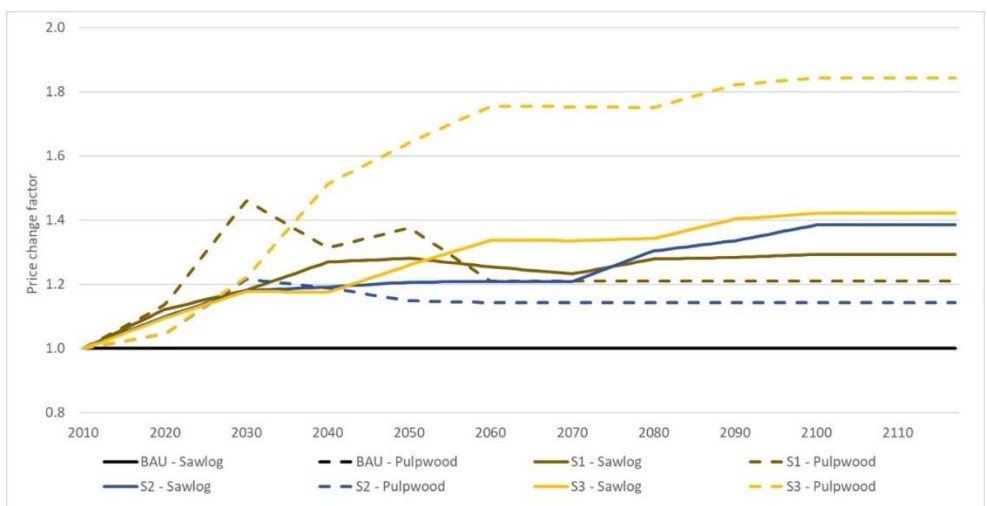

**Figure 3.** Annual dynamic price change factors used for Business as usual (BAU), S1, S2, and S3, using 2010 as start year. The BAU scenario had no dynamic price change and was included in the graph for reference.

### 2.7.3. Objective Function and Scenarios

The linear-programming objective function that was used in the Woodstock model was to maximize NPV from mill-gate timber sales while complying with forest policy and environmental protection policy. The discount rate was set at 5%, which is often used in Irish forestry [4,38,59]. Constraints were implemented to ensure that any clearfelled sites would be replanted (in accordance with forest regulation) and the total clearfell area each year was limited to no more than 300 ha, i.e., 3% of the forest area (assuming that the entire forest estate had even age-class distribution and was fellable on a 40-year rotation, 250 ha would be clearfelled in each year). This was implemented as Coillte staff, upon reviewing the results, indicated that dramatic spikes in public sector production were not feasible given their policy of ensuring an even supply of timber at the regional level. The BAU scenario and three global scenarios were run for a 100-year planning horizon, while using the Woodstock model to investigate the potential impact CC and DP had on the harvest volumes and the forest composition of the Western Peatland forests. To further investigate the impact of the two aspects of the three global scenarios, the model was only run with the CC impact but without the DP—these runs were suffixed noDP (no dynamic prices), and with only the DP but without the CC impact—these runs were suffixed noCC (no climate change). The noDP and noCC scenarios were also run for 100 years.

## 3. Results

The results are presented in three sections: (Section 3.1) change in forest composition in the global scenarios, (Section 3.2) global scenario impact on harvesting and NPV, (Section 3.3) impact on harvesting and NPV from dynamic prices and climate change separately.

### 3.1. Change in Forest Composition in the Global Scenarios

The forest composition changed over the planning horizon through the replacing of Sitka spruce and other conifer stands with lodgepole pine, by 2070 (Figure 4). There was little to no change in forest composition after year 2070. This change affected the BAU and three global scenarios: the area of lodgepole pine monocultures increased from around 26.0% in 2017 to 58.0%, 62.2%, 57.6%, and 60.0% by 2070 for BAU, S1, S2, and S3, respectively. In the scenarios in which a smaller area was converted to lodgepole pine, a larger area of Sitka spruce mixtures and other coniferous stands on blanket peat were maintained. There was a substantial increase in the total buffer zone area, mainly as aquatic buffers, which increased from 0.9% in 2017 to 5.3–6.8% of the area in 2070, depending on the scenario.

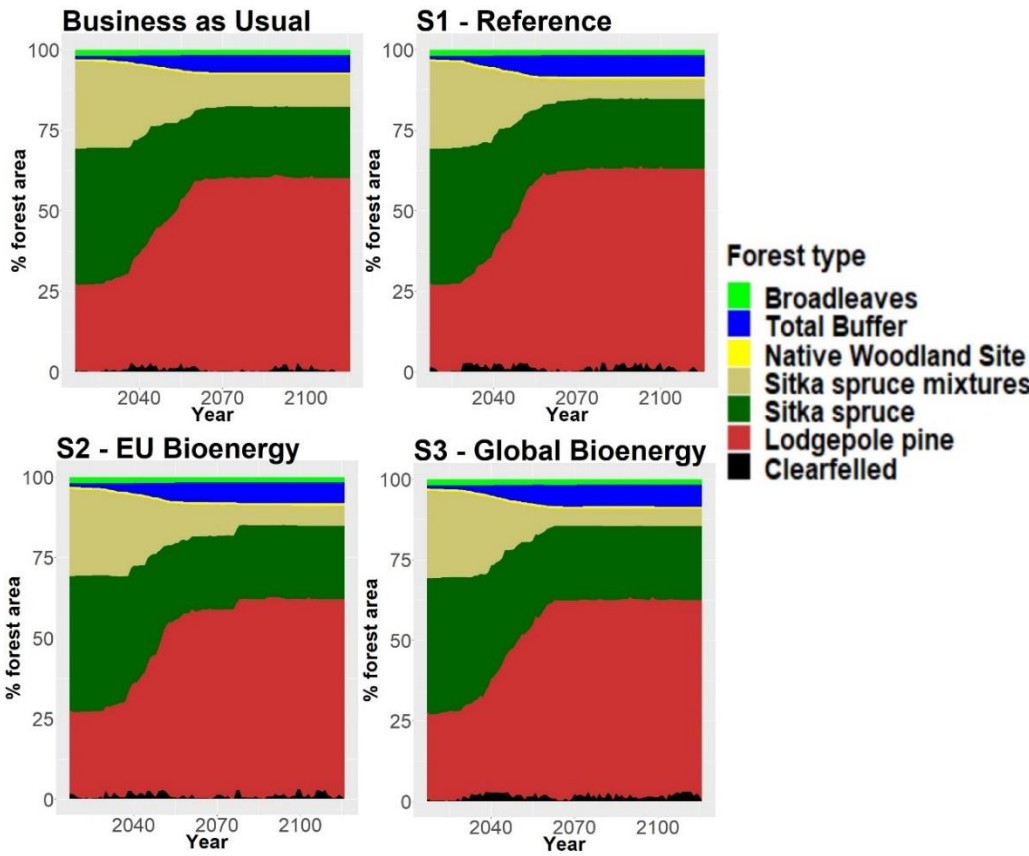

**Figure 4.** Percentage area by forest type for the four energy policy scenarios over the planning horizon. The Broadleaves group contains both privately owned broadleaves, which can be clearfelled, and Coillte broadleaves, which are managed under a continuous cover forestry regime. Total Buffer contains all freshwater pearl mussel setbacks, aquatic setbacks, and road buffers. Native Woodland Sites are mainly unmanaged native broadleaf stands that are retained for their high biodiversity values; they are separate from the 'Broadleaves' group. Sitka spruce mixtures contain all stands dominated by non-lodgepole pine conifers (including Sitka spruce), with broadleaves and/or non-lodgepole pine conifers as secondary or tertiary species. Sitka spruce and lodgepole pine refers to monoculture stands of the respective species, i.e., all the trees are the same species and same age.

*3.2. Global Scenario Impact on Harvesting and NPV*

Although the change in forest type was similar for the BAU and three global scenarios, large differences were observed in harvest area, harvest volumes, and NPV (Table 7). The NPV for the 100-year planning horizon was €16.25 M, €23.15 M, €20.55 M, and €25.71 M for the BAU, S1, S2, and S3, respectively. The development of NPV over time (i.e., the NPV in each year is the sum of all discounted costs and revenues in the preceding years and the current year, discounted to the start year 2016) shows an initial divergence between scenarios around year 2035 (Figure 5). The divergence largely stabilized by 2057, with the exception for S3 in which the NPV increased throughout the planning horizon. For the other scenarios, there was almost no increase in NPV after 2057. The total clearfell area also increased for the global scenarios when compared to the BAU, but the increase was not linearly correlated with NPV, i.e., additional clearfell area did not proportionally increase the NPV. The harvest volumes peaked in the beginning and end of the planning horizon, with a dip in the middle for all scenarios. The largest amount of total harvest volume was achieved in S3, followed by S1, S2, and lastly the BAU scenario. This was related to the DP increase for pulpwood in the global scenarios, meaning that increased prices made more forests profitable to manage for pulpwood production. The largest change in timber assortment volumes over the planning horizon was that most of the total sawlog volume was harvested in the first five decades of the model run, while pulpwood volume was harvested throughout the planning horizon and it became the dominant assortment in the second half of the planning horizon. Small volumes of sawlogs were harvested throughout the planning horizon due to the presence of mineral soils that can support a spruce crop. Standing volume increased for scenarios from around 165 $m^3$ $ha^{-1}$ in 2016 to 308 $m^3$ $ha^{-1}$, 211 $m^3$ $ha^{-1}$, 260 $m^3$ $ha^{-1}$, and 212 $m^3$ $ha^{-1}$ for BAU, S1, S2, and S3 in 2116, respectively. The difference in standing volume was due to the scenarios different harvesting levels. Although the standing volume increased overall, it slightly declined during the two large harvesting events between 2030–2050 and 2080–2016. Figure 6 presents the standing volume, harvested total, and assortment volumes.

**Table 7.** Comparison of Net Present Value (NPV) over the 100-year planning period, relative NPV, total clearfell (CF) area, relative total CF area, total extracted harvest volume, and relative total harvest volume between the BAU and the three global scenarios. Relative values are calculated using BAU as reference.

| Scenario | NPV (1000s €) | Relative NPV | CF Area (ha) | Relative CF Area | Harvest Volume (1000s $m^3$) | Relative Volume |
|---|---|---|---|---|---|---|
| BAU | 16,253 | 1.00 | 8149 | 1.00 | 3280 | 1.00 |
| S1 | 23,154 | 1.42 | 13,114 | 1.61 | 4522 | 1.38 |
| S2 | 20,554 | 1.26 | 11,380 | 1.40 | 4280 | 1.31 |
| S3 | 25,709 | 1.58 | 16,488 | 2.02 | 5277 | 1.61 |

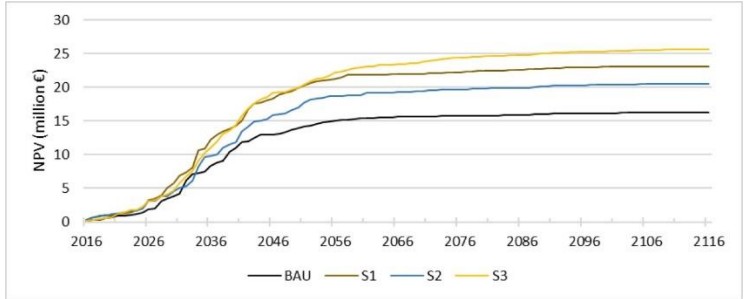

**Figure 5.** NPV development over time in the 100-year planning horizon for the BAU and three global scenarios.

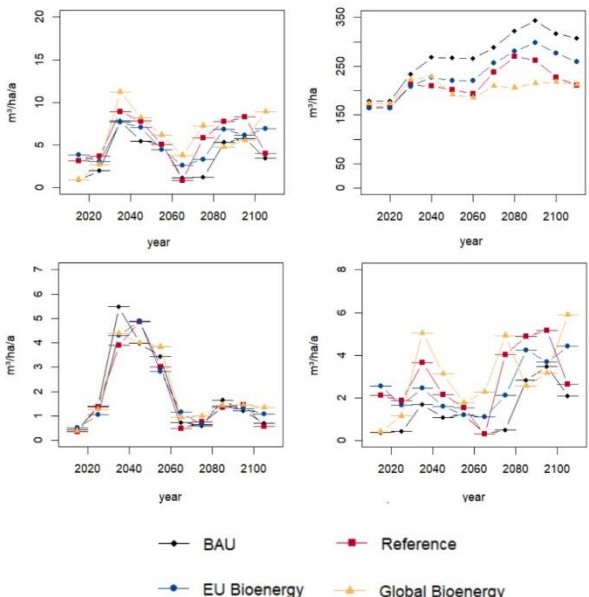

**Figure 6.** Annual harvested volume per hectare (**top-left**), standing volume (**top-right**), harvested sawlog volume (**bottom-left**) and harvested pulpwood volume (**bottom-right**) for the BAU and three global scenarios over the 100-year planning horizon. To show the trends, 10-year average values rather than actual annual volumes are presented. All values are in $m^3$ $ha^{-1}$ $year^{-1}$.

### 3.3. Impact on Harvesting and NPV from Dynamic Prices and Climate Change Separately

The differences in harvest volumes and NPV between all scenarios were more due to the DP than to CC. The noCC scenarios were more similar to the three base scenarios (i.e., S1, S2, and S3) than the noDP scenarios in terms of NPV, clearfell area, and harvested volume (Table 8). The scenarios that only incorporated CC and not DP did not achieve as high an NPV as the BAU scenario, despite lodgepole pine being predicted to grow better under all CC scenarios than in the BAU scenario. The CC factors reduced the growth for commercially valuable conifers (i.e., spruces, firs, and Doulas fir). Thus, the higher the CC effect, the less volume could be harvested from these species, reducing NPV. Two of the noDP scenarios, utilizing only CC, had lower NPV than the corresponding BAU scenario. With an NPV of €16.24 M, S3_noDP almost managed to produce an equal NPV to the BAU scenario, but only by harvesting 35% more area and 20% more net volume, with most of this increase consisting of pulpwood.

**Table 8.** Comparison between NPV over the 100-year planning period, relative NPV, total clearfell (CF) area, relative CF area, total extracted harvest volume, and relative harvest volume for the BAU, S1, S2, S3, noCC and noDP scenarios. Relative values are calculated using the BAU values as reference.

| Scenario | NPV (1000s €) | Relative NPV | CF Area (ha) | Relative CF Area | Harvest Volume (1000s m³) | Relative Volume |
|---|---|---|---|---|---|---|
| BAU | 16,253 | 1.00 | 8149 | 1.00 | 3280 | 1.00 |
| S1 | 23,154 | 1.42 | 13,114 | 1.61 | 4522 | 1.38 |
| S2 | 20,554 | 1.26 | 11,380 | 1.40 | 4280 | 1.31 |
| S3 | 25,709 | 1.58 | 16,488 | 2.02 | 5277 | 1.61 |
| S1_noCC | 24,858 | 1.53 | 12,727 | 1.56 | 4765 | 1.45 |
| S2_noCC | 21,873 | 1.35 | 10,750 | 1.32 | 4247 | 1.29 |
| S3_noCC | 25,357 | 1.56 | 14,623 | 1.79 | 5220 | 1.59 |
| S1_noDP | 15,078 | 0.93 | 9897 | 1.21 | 3564 | 1.09 |
| S2_noDP | 15,246 | 0.94 | 8771 | 1.08 | 3325 | 1.01 |
| S3_noDP | 16,239 | 1.00 | 11,004 | 1.35 | 3940 | 1.20 |

## 4. Discussion

Western Peatland forests, such as those in the Barony of Moycullen, are undergoing large changes in forest composition independently of the current and future climate and the demand for wood fibre. The desire for public forests in Ireland to become certified will mean an increase in lodgepole pine area, due to peat sites not being reforested using fertiliser. Management will likely not revert to fertilisation in the CSA due to the environmental protection status of adjacent land and the Owenriff catchment freshwater pearl mussel population's sensitivity to eutrophication and siltation. The other large-scale forest composition change was the establishment of aquatic buffer zones. The forests were planted right up to the water body before Irish forestry started adapting towards SFM and increased its environmental consideration in 1996 [60]. Current forest practice is to parcel off the area adjacent to waterbodies and establish buffer zones during subsequent forest management actions. Therefore, these major changes in forest composition were due to the implementation of these new environmental policies and regulations, and not due to the climate and price changes of the global scenarios. The change in forest composition, from a Sitka spruce to a lodgepole pine dominant landscape, was not initiated until 2029, year 13 in the planning horizon. There was little harvesting in the early years, because the linear programming optimisation delayed harvesting to utilise higher assortment prices in the future. The early harvesting was primarily of lodgepole pine stands on blanket peat or spruce stands on mineral soil, these were reforested with the same species, which did not cause any change in forest composition. The low harvesting rates are consistent with the current Coillte management plan for the area [61]. The dip in harvest volume in the middle of the planning horizon was due to the forest in the CSA having an uneven age-class distribution, with 75.7% of the forest area being older than 20 years at the start year. After these stands were harvested, most of the forest consisted of juvenile stands, causing the dip in harvesting.

The overall results show an increase in NPV from the management of the Western Peatland forests under the three global scenarios when compared to the BAU scenario. However, it is important to emphasise that the NPV increases were due to increasing assortment prices rather than the biophysical effects of CC. CC is likely to reduce the revenue of forestry in Western Ireland [30], especially from sawlog-producing conifer stands. Spruces, firs, and Douglas firs in the CSA are projected to grow 9%–18% slower in the future under the A2 CC scenario, reducing the potential revenue from forestry. These predicted growth reductions of Sitka spruce in the west of Ireland are similar to the results of another Irish study that used Climadapt to analyse growth trends over the whole country [30], indicating that CC will negatively affect NPV for most Irish forestry. The higher demand for wood fibre in the expanding bioeconomy can mobilize biomass from marginal forests that would not be harvested under current conditions. Blanket bog lodgepole pine forests, seen as financial loss-making today [4], could be more valuable for foresters in the future if the pulpwood prices increase in the long-term. Although based on the DP used in this study, they will not become as profitable as coniferous plantations that are grown for sawlogs. The reduction in harvested sawlog volumes in the second half of the planning horizon for all scenarios was due to lodgepole pine only being utilised for pulpwood, based on its low quality (e.g., lack of straightness, excessive knots, etc.). Low-value lodgepole pine and the diminishing returns of discounting revenues from far into the future were the reasons why the NPV stagnated around the year 2057. NPV in the S3 scenario grew the most after 2057, compared to the other scenarios, but it was also the only scenario where the DP increased throughout the planning horizon. The reason that additional clearfelling area did not increase NPV linearly was partly due to diminishing returns of discounting future revenues. Additionally, any additional harvesting in S1, S2, and S3 when compared to the BAU scenario was of blanket bog grown lodgepole pine. The mill gate prices for Sitka spruce and lodgepole pine at average tree size 0.5 $m^3$ is €33.7 $m^{-3}$ and €7.1 $m^{-3}$, respectively, and at average tree size of 1 $m^3$ it is €42.1 $m^{-3}$ and €10.7 $m^{-3}$, respectively. These are present day prices, without discounting and without consideration reforestation costs.

The NPV reduction, when only modelling CC, is consistent with the findings of Keenan et al. [31]. Their study modelled CC on a very similar forest in the same part of Ireland. However, they

implemented the entire growth impact of CC in year 2050 and 2080 (depending on the CC scenario), and it was implemented by changing the species specific YC, rather than scaling the volume growth with an annual growth impact factor, as was done in this study. Another study used national forest inventory soil data, rather than the default Climadapt low-resolution soil maps, to analyse the CC impact on Sitka spruce productivity in the south-east of Ireland [30]. They predicted the same baseline productivity and both projections showed reduced productivity in the long-term. However, by using more accurate soil data, Sitka spruce productivity was predicted to be one to two YCs higher for the mid-term and long-term projections when compared to those using the default soil map [30]. Regardless of how and at which spatial scale CC is modelled for the Western peatland forests, the results point towards a negative impact on NPV due to the reduced growth of tree species utilised for sawlog production. However, the impact of CC on forest productivity widely differs across Europe. Projections of CC impact on stand productivity indicated increased stand productivity in an area of eastern Germany; this caused an increase in NPV from timber sales [62]. Increased forest productivity is also expected in temperate-oceanic areas of northern France [63] and in boreal forests in Sweden [64] and Finland [65,66]. Decreases in forest productivity are expected in south-western France [63], while Norway spruce is expected to become an unsuitable timber species in lowland Austria in the future [67]. Increased temperatures and reduced precipitation resulting from CC is expected to reduce wood production in the Mediterranean area [68]. Thus, the spatial scale at which the CC impacts are calculated for FMDSS implementation will affect the accurate forecasting of the forest conditions, influencing the adjustments to management that can be made. Even at the CSA scale, the Climadapt predictions differed slightly over the CSA, so there might be a research opportunity to implement CC at different spatial scales to evaluate the effect on the resulting best adaptive forest management. Additionally, the data in Climadapt is scaled from a low-resolution dataset [14]; higher resolution CC data would be useful in improving the accuracy and precision of forecasting CC impacts on forest productivity at the forest or landscape level.

Due to current forest policy and environmental regulations, aerial fertilisation is no longer permitted when planting trees in the CSA, and there is not enough manpower for manual fertilisation [44]. Thus, the area of planted Sitka spruce has declined in recent years and, apart from a few sites with shallow blanket peat, the spruce trees are no longer planted on blanket peat in the CSA [49]. Thus, the NPV reduction was an effect of having a legacy forest at the start of the planning horizon; a forest that would not be established today. Fertilisation and spruce planting are still common management practices in Irish blanket peat forests with fewer environmental constraints than the CSA; however, increasing environmental regulation could cause blanket bog fertilisation to entirely cease in the future. Coillte has dual-certification and fertilise several of their plantations, but Irish certification rules for Forest Stewardship Council and Programme for the Endorsement of Forest Certification are to limit the use of fertiliser that is used in forestry and only use it when necessary to ensure canopy closure [69,70]. Therefore, it is likely that the species choice will be re-evaluated for poorly performing sites when it is time to plant the second or third generation of trees. The alternative management of peatland forests might also be attractive to better utilise the sites and provide other ESs (such as carbon sequestration, cultural services, and biodiversity), instead of low value pulpwood [71]. These options may include long-term retention of forest, restoration of natural bog habitat through rewetting, natural regeneration, retention of unplanted areas, planting with native species, and planting with lodgepole pine at low stocking levels [6,72].

The GROWFOR yield tables that were used were primarily developed using sites with mineral soils; only one of the 14 Sitka spruce sample sites was located on blanket peat and none of the other modelled conifers utilised for sawlog production had any sample sites that were located on blanket peat [73]. Due to the poor biophysical growing conditions in the Western peatlands, there was likely an overestimation of the merchantable volume that could be extracted from blanket peat stands. Based on observations that were made by Coillte staff, many blanket peat stands do not have full stocking due to high seedling mortality as a result of poor site conditions [74]. Many blanket peat stands in the

forest inventory data were indeed not fully stocked and these stands perform differently, depending on peat depth [45]. Accurate data on peat depth was not available and there was no consistency in the stocking rate of blanket peat stands, thus it was impossible to appropriately model the success rate of future reforestation in the Woodstock model. Some stands, including fertilised spruce and unfertilised lodgepole pine, go into check about a decade after planting. Check is a condition where the trees stop growing and foliage turns yellow due to insufficient mineral nutrients in the soil. Peat depth is likely a good predictor of YC [75] and it could be an indicator of high seedling mortality and the possibility of the stand going into check. However, there were no reliable data available to incorporate check and plantation failure based on site characteristics in the model.

The impact of CC was not focused on excluding unsuitable species in the model. However, a more suitable species would be chosen if a species suffered severe growth reductions as a result of CC. Productivity reduction alone does not necessarily mean a species should be changed. If Sitka spruce YC 14 suffers a 15% growth reduction, then it will still grow better than lodgepole pine YC 10 on the same site that benefits from a 10% growth increase, as Sitka spruce would have a YC of 11.9 and lodgepole pine a YC of 11.0. However, depending on the DP it might be favourable to plant one species over the other. The model did not include the frequency of current and future pest and disease outbreaks, as relevant information is lacking [76]. Irish forests have been relatively safe from forest pests in the past [77] and forest health is currently considered to be good [78]. Homogenous forests are generally more susceptible to pest and disease spread [79], which could have devastating effects on the CSA forest if an invasive species arrives. Given that the increased spreading of invasive forest pests and diseases is already an observed outcome of a changing climate and increased global trade [80], the risk to Ireland's homogenous peatland forests is something that needs to be considered with a greater urgency, and research is required to quantify the risks and their consequences.

Uncertainties arise around how much foresters will know about future timber prices and tree growth under CC, and how management should be adapted to meet these challenges. It is unlikely that foresters will have full knowledge of these global impacts and will be able to optimize their forest management accordingly. However, based on corporate market analyses and large quantities of research on CC, foresters will probably be able to adapt and adjust to a high degree. This study focused on the impacts on harvest volumes and NPV. However, CC and increased harvesting resulting from higher assortment prices are likely to impact other benefits from forests, such as biodiversity, water quality, and carbon stocks. Predicting the impact global scenarios will have on these other ESs is important to assess the trade-offs from increased harvesting rates. Compliance with SFM means biodiversity and social values must also be considered, besides economic values. This is currently the focus of a project that builds on the results that are presented in this paper.

## 5. Conclusions

With reliable data on future conditions, modelling the impact of CC and DP on forest management can be implemented in Remsoft's Woodstock, without having to change yield tables. The results indicate that higher demand for wood fibre will offset the negative effects of climate change in the CSA. Climate change will negatively affect the growth of conifer species that are utilised for sawlog production in the Western peatland forests in Ireland, resulting in a reduction in NPV as compared to current growing conditions. Based on this study, it is recommended that foresters incorporate global changes in their long-term management plans to mitigate the negative effects that non-adaptive management decisions can have on their forest enterprises.

**Author Contributions:** Conceptualization, M.N.; Methodology, A.L., E.C. and M.N.; Software, A.L. and E.C.; Validation, A.L., E.C. and M.N.; Formal analysis, A.L.; Investigation, A.L. and E.C.; Resources, M.N.; Data curation, A.L.; Writing—Original draft preparation, A.L.; Writing—Review and editing, A.L., E.C. and M.N.; Visualization, A.L.; Supervision, E.C. and M.N.; Project administration, M.N.; Funding acquisition, M.N.

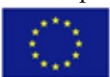 **Funding:** This project has received funding from the European Union's Horizon 2020 research and innovation programme under grant agreement No 676754.

**Acknowledgments:** We would like to thank Kevin Black for applying the Climadapt calculations in our land-use layer. We would like to thank the Coillte staff who contributed with forest data, growth and yield tables, more realistic management prescriptions and their associated costs: Liam Malone, John Landy, Paul Ruane, Tony Clarke and Frank Flanagan. We would also like to thank Remsoft for having truly exceptional customer support staff.

**Conflicts of Interest:** The authors declare no conflict of interest. The funders had no role in the design of the study; in the collection, analyses, or interpretation of data; in the writing of the manuscript, or in the decision to publish the results.

**Disclaimer**: Responsibility for the information and views set out in this article/publication lies entirely with the authors.

**Appendix A**

**Table A1.** Costs (€ ha$^{-1}$) for individual silviculture actions, based on conifer and broadleaf forests. Source: Coillte.

| Action | Conifer | Broadleaf |
|---|---|---|
| Mounding | 650 | 650 |
| Fencing | 89 | 204 |
| Planting | 1500 | 2077 |
| Weevil control & Vegetation control | 300 | 300 |
| Inspection | 50 | 50 |

**Table A2.** Value of standing volume (in € m$^{-3}$) used in the model, based on average tree size for conifers (excluding lodgepole pine), and fixed values for broadleaves and lodgepole pine. Source: Teagasc [58] and Coillte.

| | Standing Volume Value | | |
|---|---|---|---|
| Average Tree Size m$^3$ | Conifer (Excluding Lodgepole) | Broadleaves | Lodgepole Pine |
| 0.001–0.174 | 10.42 | 41.00 | 26.00 |
| 0.175–0.274 | 28.20 | 41.00 | 26.00 |
| 0.275–0.374 | 38.00 | 41.00 | 26.00 |
| 0.375–0.474 | 41.96 | 41.00 | 26.00 |
| 0.475–0.599 | 45.89 | 41.00 | 26.00 |
| 0.600–0.799 | 49.05 | 41.00 | 26.00 |
| 0.800–0.999 | 50.66 | 41.00 | 26.00 |
| >1.000 | 52.17 | 41.00 | 26.00 |

**Table A3.** Felling and extraction costs (in € m$^{-3}$) used in the model, based on average tree size, harvesting operation and species. Source: Coillte.

| | Felling and Extraction | | |
|---|---|---|---|
| | Conifer (Excluding Lodgepole) and Broadleaves | | Lodgepole |
| Average Tree Size m$^3$ | Clearfelling | Thinning | Clearfelling |
| 0.001–0.174 | 13.35 | 20.17 | 16.55 |
| 0.175–0.274 | 11.42 | 16.31 | 14.34 |
| 0.275–0.374 | 15.56 | 14.30 | 12.21 |
| 0.375–0.474 | 9.83 | 13.52 | 11.72 |
| 0.475–0.599 | 9.22 | 12.60 | 10.86 |
| 0.600–0.799 | 8.19 | 10.73 | 9.62 |
| 0.800–0.999 | 7.15 | 8.90 | 7.31 |
| >1.000 | 7.15 | 8.90 | 7.31 |

**Table A4.** Haulage cost (in € m$^{-3}$) used in the model, based on species assortment and transport distance to suitable processing mills.

|  | Alder | Ash | Beech | Birch | Douglas Fir | Larch | Lodgepole Pine | Maple | Norway Spruce | Oak | Scots Pine | Sitka Spruce |
|---|---|---|---|---|---|---|---|---|---|---|---|---|
| Haulage cost | 3.25 | 4.11 | 3.89 | 3.68 | 3.25 | 3.32 | 7.99 | 3.75 | 2.96 | 4.18 | 3.03 | 2.96 |

**Table A5.** Additional costs (in € m$^{-3}$) for harvest activities in specific areas. Source: Coillte.

| Environmental Designation | Cost |
|---|---|
| Harvesting in special areas of conservation or freshwater pearl mussel catchment | 0.20 |
| Harvesting in special protection areas, national heritage areas or proposed national heritage areas | 0.10 |
| Harvesting on peat soils or in buffer | 2.60 |

**Table A6.** Price change factors used in the three scenarios. Although the table only contains values for every decade, the price change factor was linearly interpolated and implemented on an annual basis. For example, a factor of 1.00 corresponds to no price change compared to the 2010 initial price and a factor of 1.10 corresponds to a 10% price increase.

| Dynamic Price Change Factors | | | | | |
|---|---|---|---|---|---|
| S1 | | S2 | | S3 | |
| Year | Sawlog | Pulpwood | Sawlog | Pulpwood | Sawlog | Pulpwood |
|---|---|---|---|---|---|---|
| 2010 | 1.00 | 1.00 | 1.00 | 1.00 | 1.00 | 1.00 |
| 2016 | 1.07 | 1.08 | 1.06 | 1.03 | 1.06 | 1.03 |
| 2020 | 1.12 | 1.14 | 1.10 | 1.05 | 1.09 | 1.05 |
| 2030 | 1.18 | 1.46 | 1.18 | 1.22 | 1.18 | 1.22 |
| 2040 | 1.27 | 1.31 | 1.19 | 1.19 | 1.18 | 1.51 |
| 2050 | 1.28 | 1.38 | 1.21 | 1.15 | 1.26 | 1.64 |
| 2060 | 1.25 | 1.21 | 1.21 | 1.14 | 1.34 | 1.75 |
| 2070 | 1.23 | 1.21 | 1.21 | 1.14 | 1.34 | 1.75 |
| 2080 | 1.28 | 1.21 | 1.30 | 1.14 | 1.34 | 1.75 |
| 2090 | 1.28 | 1.21 | 1.34 | 1.14 | 1.40 | 1.82 |
| 2100 | 1.29 | 1.21 | 1.38 | 1.14 | 1.42 | 1.84 |
| 2116 | 1.29 | 1.21 | 1.38 | 1.14 | 1.42 | 1.84 |

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
