# Peer review of "Implementing Climate Change and Associated Future Timber Price Trends in a Decision Support System Designed for Irish Forest Management and Applied to Ireland’s Western Peatland Forests"

_forests, doi:10.3390/f10030270_

Round 1
Reviewer 1 Report
Thank you for letting me review this interesting manuscript.
While I think that the manuscript is of general interest, I think there are serious flaws that need to be dealt with before it can be considered for publication.
General comments:
The methods section is very detailed, so it is difficult to filter the information relevant to understand and interpret the results. There are 12 tables in total, which I think is too much. Please consider summarizing some of the information, or to move some of it to an appendix if the journal allows this (e.g. Tables 6, 9 and 10). Figure 3 and Table 12 show essentially the same information, so I think the table is not needed.
Also, the text needs careful editing. On p8, the text is interrupted, and on p13, there is a graph without caption in the middle of the text, by mistake, I assume. This is confusing for the reader.
Specific comments:
Maybe I did not find it, but you don’t seem to mention the length of simulation period in the methods, is it 100 years?
P2L58-76: you often use the word ‘will’ for describing some future trend. Consider using some other word(s), such as are expected to, unless you are completely certain that something will happen (there is always some degree of uncertainty in projections).
P3L110: you write that Woodstock is a tactical planner – but you use it for strategic planning? This is confusing, you should make clear here why you think it is applicable for strategic planning as well.
P3L125: In the beginning of the introduction, you write that climate changes leads to increased productivity for most of the country. Here you say there is a rather large growth reduction – why?
P3L130-135: I don’t really get what you want to say here – consider rephrasing. I think a reader needs more information to understand these statements.
P3L137: You do not mention dynamic timber prices in the text leading up to your objectives. I think you need to write some text about why they are important and relevant enough to be such a big part of your study?
P4, CSA: How representative is your CSA of Irish forests? Why did you choose this particular CSA?
P4L148: What is priority 8? Is it because of the mussel that you need the aquatic buffers, or is that something you do in general?
Table 1: temp and rainfall ranges: what are these ranges, min – max? For which years? Please clarify. Species cover – is is % of volume or area?, Ownership – What are these grants? Is it important for the reader to understand the difference between grant-aided and non-grant aided? If so, explain, if not, you could probably summarize the two? Productivity – is it in %?, Age class distribution – is it by area or volume?, soils, forest% - do you mean the forest in the CSA? You don’t show any info on standing volume, might be valuable to add (maybe also to results).
P6L181: you mention ecosystem services here for the first time, and that you will write about how you included them, but as far as I can see, you don’t really come back to that in the text.
P7L187ff: naming of the model and use in different projects is less relevant to the reader compared to the changes made in it, so focus on that. I think you need be more specific on the optimization and constraints, this is quite critical to understand your results, I think.
Also, did you get the info for private forests also from Coillte? How was the forest information gathered?
P7L201: ‘linked database’ – linked to what?, ‘imported shapefile’ – imported where? Please be more specific.
P7 (input data): Consider giving the source for the forest input data when you describe it for the first time, i.e. in Table 1.
P12L287: Was the future YC really only projected for one single year, i.e. using the climate information from the year 2080? Given that there supposedly is a lot of inter-annual variation, this does not seem to be reliable enough in my view. Or was some kind of average climate over several years used, then please state that in the text.
P7L218 What is SS YC? I don’t see it explained anywhere.
P8, Dynamic assortment prices: I think you need to give some more information about the GLOBIOM model – what it does, based on which data, etc. To my knowledge, the prices from GLOBIOM come together with harvest volumes, but you did not use those, right? You should probably also mention that GLOBIOM does not model climate change impacts on forest productivity.
P9, Table 5: You don’t mention Natura 2000 in the description of the CSA, so it comes a bit as a surprise here.
P12L287: What RCP does A2 compare to?
P13L315: In the graph it looks like sawlogs prices are highest in S3?
P15L347ff: To me, forest structure is much more than tree species distribution. Consider writing about tree species distribution instead.
Figures 3, 5 and 6: would be helpful if all use the same colour coding.
Figure 4: Colours difficult to differentiate, can you try to increase contrast? How do you define monoculture?
Figure 5: I think you need to explain cumulative NPV – I don’t think it is a very commonly used concept. At least I have not seen it before. You may also consider changing the y-axis to million €.
Table 13: what are the area and volume numbers, sums for the whole 100 year period?
P18L390 Difference compared to what, between scenarios?
P18L39: do you mean difference between scenarios?
P18L392ff: Why is that? You did not use dynamic prices in BAU, did you, so this seems counter-intuitive and needs an explanation.
P18L405: I may have missed it, but I didn’t see you mention certification issues specifically in the methods. Should be mentioned there.
Table 14: consider showing NPV as 1000 Eur to increase readability. Also harvest volume could be shown in 1000 m3. Why is the relative NPV for S3_noDP 1.09 when the absolute value is in fact smaller than BAU?
P18L406: I don’t think you should bring up new results in the discussion section. If you want to make this point, you need to mention the results of these extra runs already in the methods & results.
P19L414: This information needs to be mentioned already in the methods, I think.
P19L422: harvesting and reforestation, even with same tree species, can be considered to be quite a big change in forest structure. I guess you mean tree species distribution?
P19L424: ‘normal age-class distribution’ – do you mean even, i.e. that your input data had an uneven age-class distribution?
P19L431: is your CSA representative of whole Ireland so that you can make this statement confidently?
P19L448ff: for which scenario and which time point are these prices?
P19L457-459: which study do you mean here, the Keenan et al study or yours?
P20L460: increase compared to what?
P21L539: Reduction compared to what?
Author Response
Response to Reviewer 1 comments (Round 1)
Authors note: Answers and the corresponding manuscript edits (i.e. tracked changes) in to Reviewer 1 comments are marked R1 P X (Reviewer 1 Point X) in the revised manuscript. The below answers will have a corresponding comment in the revised manuscript and the cover letter detailing all changes made.
Thank you for letting me review this interesting manuscript.
While I think that the manuscript is of general interest, I think there are serious flaws that need to be dealt with before it can be considered for publication.
General comments:
The methods section is very detailed, so it is difficult to filter the information relevant to understand and interpret the results. There are 12 tables in total, which I think is too much. Please consider summarizing some of the information, or to move some of it to an appendix if the journal allows this (e.g. Tables 6, 9 and 10). Figure 3 and Table 12 show essentially the same information, so I think the table is not needed.
R1 general comment If you think it would make the text easier to read, I agree that table 6, 9 and 10 could go to supplementary material. Table 7 and 8 could also be moved to supplementary to keep all the costs and revenues in one place.
I would suggest moving table 12 to supplementary material, rather than removing it completely. I’d like to supply the actual numbers, in case anyone would like to replicate the study.
These changes have been made, so table names are changed
Also, the text needs careful editing. On p8, the text is interrupted, and on p13, there is a graph without caption in the middle of the text, by mistake, I assume. This is confusing for the reader.
P8: Does this refer to the interruption between line 220-221? This has been fixed.
P13: I think this graph got added by mistake when I cross-referenced the figure number. It has been removed as it is a duplicate of the graph below.
Specific comments:
Maybe I did not find it, but you don’t seem to mention the length of simulation period in the methods, is it 100 years?
R1P1 It is mentioned in the results, but I’ve now added it to Materials and methods – Objective function and scenarios as it fits best here.
P2L58-76: you often use the word ‘will’ for describing some future trend. Consider using some other word(s), such as are expected to, unless you are completely certain that something will happen (there is always some degree of uncertainty in projections).
R1P2 The writing of this section has been changed to express less certainty about the future.
P3L110: you write that Woodstock is a tactical planner – but you use it for strategic planning? This is confusing, you should make clear here why you think it is applicable for strategic planning as well.
R1P3 This is purely a mistake. Woodstock is indeed the strategic planner.
P3L125: In the beginning of the introduction, you write that climate changes leads to increased productivity for most of the country. Here you say there is a rather large growth reduction – why?
R1P4 Thanks for pointing this out. I’ve rewritten the earlier sentence on L64 to make it clearer. The previous sentence was meant to summarise climate change research that increased CO2 and temperature alone generally increase growth. After this I point out that due to changed precipitation the forest soils in Ireland are likely to limit growth for Sitka sprue through drought and excessive wetness (depending on geographical region).
I’ve added a sentence to bridges these statement/
From:
Increased CO2 levels and temperature will both stimulate higher biomass productivity and thus forest growth, for most of the country [9,11].
To:
Increased CO2 levels and temperatures would generally be expected to result in higher biomass productivity, and thus forest growth, for most of the country[9,11]. However, when considering future precipitation patterns, soil types, and species response it is not likely that all species will experience increased growth rates in the future.
P3L130-135: I don’t really get what you want to say here – consider rephrasing. I think a reader needs more information to understand these statements.
R1P5 Changed from:
Another Irish study modelled a high-emission CC scenario in Woodstock by implementing Climadapt’s A2 scenario prediction on future YC in the years 2050 and 2080 to evaluate the impact on forestry in Western Ireland [32]. The NPV of forestry was reduced more in the 2050 YC prediction than in the 2080 YC prediction. The 2080 CC impact reduced NPV and had more of a negative effect on growth. However, earlier implementation of CC’s negative impact on the growth of commercially valuable species had a stronger impact on the NPV.
To:
Another Irish study implemented Climadapt’s A2 CC scenario future YC predictions in Woodstock to evaluate CC long-term impact on forestry in western Ireland [32]. Their model was run three times: without YC change, YC changed in year 2050 (future climate based on the period 2020-2050), and YC changed in year 2080 (future climate based on the period 2050-2080). The NPV of forestry was reduced when changing the YC, but the NPV was reduced more in the 2050 YC prediction than in the 2080 YC prediction. The 2080 YC change had more of a negative effect on the productivity, but earlier implementation of CC’s negative impact on the growth of commercially valuable species had a stronger negative impact on the NPV.
P3L137: You do not mention dynamic timber prices in the text leading up to your objectives. I think you need to write some text about why they are important and relevant enough to be such a big part of your study?
R1P6 Added a paragraph in the introduction that mentions adverse impact of climate change, mitigation strategies, and forest owner response to changed prices.
P4, CSA: How representative is your CSA of Irish forests? Why did you choose this particular CSA?
R1P7 On L146-150 some reasons are laid out: high recreation pressure, lots of adjacent protected area, and proximity to freshwater pearl mussel habitat.
Few forest landscapes in Ireland would be representative for the whole country, but I added an explanation that it is representative for forests of Western Ireland.
It was chosen for a research project because we wanted to investigate ecosystem services trade-off and the potential of alternative forest management on a blanket peat forest landscape that has high recreation pressure and lots of valuable aquatic and wildlife habitat. These research points will be discussed in another paper.
P4L148: What is priority 8? Is it because of the mussel that you need the aquatic buffers, or is that something you do in general?
R1P8 Added sentence to explain what priority 8 are:
These priority catchments hold 80% of Ireland’s FPM numbers and are important for the species long-term survival.
Buffers are necessary regardless if mussels are present or not. Adjacency to freshwater pearl mussel catchment requires wider buffers.
Table 1: temp and rainfall ranges: what are these ranges, min – max? For which years? Please clarify. Species cover – is is % of volume or area?, Ownership – What are these grants? Is it important for the reader to understand the difference between grant-aided and non-grant aided? If so, explain, if not, you could probably summarize the two? Productivity – is it in %?, Age class distribution – is it by area or volume?, soils, forest% - do you mean the forest in the CSA? You don’t show any info on standing volume, might be valuable to add (maybe also to results).
R1P9 Climate data years added (i.e. 1981-2010). Temperature is given in the mean range for the month, this has been added to the table. Rainfall is the mean for any year.
Species cover is by area, this has been clarified.
Ownership – private groups have been merged, the results do not differentiate between the grant and non-grant aided and neither does the forest data.
Productivity header changed to: Productivity (YC) by stocked forest area (%).
Age class distribution by area, not volume, this has been clarified.
Soils, and elevation data, changed to “In forest” and “In CSA” to clarify that it refers to “Soils in forest, soils in CSA, etc”.
Average standing volume added to table 1 and to results.
P6L181: you mention ecosystem services here for the first time, and that you will write about how you included them, but as far as I can see, you don’t really come back to that in the text.
R1P10 Removed the mention of ESs. The Woodstock model accounts for ES, and I plan to publish a paper on this in the future. However, there it does not fit this paper. To better match the paper, this sentence now reads:
The remainder of this Material and Methods section focuses on combining the GIS data to build a forest inventory shapefile appropriate for integrating into Woodstock for strategic level planning, the growth and yield tables that were used, the eligibility for forest management prescriptions, including prescription costs and timber prices, and the implementation of the global scenarios.
P7L187ff: naming of the model and use in different projects is less relevant to the reader compared to the changes made in it, so focus on that. I think you need be more specific on the optimization and constraints, this is quite critical to understand your results, I think.
Also, did you get the info for private forests also from Coillte? How was the forest information gathered?
R1P11 I’m inclided to mention Coillte and reference the previous model. I could however remove mentions of the research projects.
The sentence about optimization and constraints can be removed as there is a section dedicated to it Objective function and scenarios. Removing this sentence reduces confusion and makes the paper less repetative.
The private forests were sources from the Irish Forest service, listed in table 2. Forest informtion (both Coillte and Forest service) are based on what is in their GIS inventory – mainly data on species, age, site productivity, managmeent . With this information we can assign a growth table to each stand and find out its height, diameter, volume etc.
This section now reads:
Coillte collaboration and UCD Woodstock model
Coillte (the Irish state forestry board, who manage Ireland’s publicly-owned forests) provided their Woodstock model to University College Dublin in February 2012 to be used in a research project [25]. Since May 2016, this Woodstock model [34] has undergone a material development phase for use in this study. Changes to the model include updated costs, revenues, growth and yield tables, policy rules, and the implementation of global climate and price change scenarios.
P7L201: ‘linked database’ – linked to what?, ‘imported shapefile’ – imported where? Please be more specific.
R1P12 This sentence changed to:
The ESRI shapefile containing spatial location of forest stands and the linked database with site information and forest inventory data was produced using ArcGIS 10.4.
P7 (input data): Consider giving the source for the forest input data when you describe it for the first time, i.e. in Table 1.
R1P13 Added a reference to table 2 in the table 1 caption.
P12L287: Was the future YC really only projected for one single year, i.e. using the climate information from the year 2080? Given that there supposedly is a lot of inter-annual variation, this does not seem to be reliable enough in my view. Or was some kind of average climate over several years used, then please state that in the text.
R1P14 The 2080 YC uses the projected average, climate for the period 2050-2080. This has been clarified in the text.
P7L218 What is SS YC? I don’t see it explained anywhere.
R1P15 Sitka spruce YC. The abbreviation has been removed and the wording should now be easier to understand.
P8, Dynamic assortment prices: I think you need to give some more information about the GLOBIOM model – what it does, based on which data, etc. To my knowledge, the prices from GLOBIOM come together with harvest volumes, but you did not use those, right? You should probably also mention that GLOBIOM does not model climate change impacts on forest productivity.
R1P16 Added more information about GLOBIOM in M&M>Global scenarios:
GLOBIOM computes the market equilibrium for agriculture, forestry and bioenergy, based on land-use competition, population dynamics, global trade, and policies. The model includes accounting of greenhouse gas emissions and can, as an example, be used to analyse how global development and policy scenarios will affect greenhouse gas emissions in the future. Although GLOBIOM incorporates agricultural adaptation to CC, GLOBIOM did not change forest productivity as a result of CC when producing the dynamic timber assortment prices for this assessment. GLOBIOM provided data on average global temperature increases for each scenario which was used to find a corresponding CC scenario in Climadapt for changing forest productivity
Wrote about the harvest volume under M&M>Global scenarios>Dynamic assortment prices:
The DP were produced by IIASA, using GLOBIOM and were based on external projections of wood demand in Ireland. The wood demand barely increased for S1 but the increases for S2 and S3 were in line with national projections of future wood harvest [59]. The national wood harvest increases rely on continued afforestation and maturing of recently afforested private forests, however, there has been very little afforestation in the CSA recently and most sites are not eligible for afforestation. Thus, achieving these increases in harvested wood volumes was not explicitly used as a constraint in the Woodstock model.
P9, Table 5: You don’t mention Natura 2000 in the description of the CSA, so it comes a bit as a surprise here.
R1P17 I’ve included a mention on the Natura 2000 areas in the CSA description.
P12L287: What RCP does A2 compare to?
R1P18 RCP8.5, this has been clarified in the text.
P13L315: In the graph it looks like sawlogs prices are highest in S3?
R1P19 This must be a typographical error, it has been fixed, and the section has been rewritten.
P15L347ff: To me, forest structure is much more than tree species distribution. Consider writing about tree species distribution instead.
R1P20 The other reviewer commented on this too and suggested I use forest composition. Is this Ok? The results and graphs include things as clearfelled area, bufferzone, native woodland sites, so it’s not strictly about tree species.
>>Manuscript changes implemented under Reviewer 2 comment R2P12<<
Figures 3, 5 and 6: would be helpful if all use the same colour coding.
R1P21 Changed for figure 3 and 6 colours, colour scheme of figure 5 was used.
Figure 4: Colours difficult to differentiate, can you try to increase contrast? How do you define monoculture?
R1P22 I’ll give it a go to increase contrast. Maybe add some other colours that are easier differentiated.
Monoculture defined: i.e. all trees are the same species and same age.
Figure 5: I think you need to explain cumulative NPV – I don’t think it is a very commonly used concept. At least I have not seen it before. You may also consider changing the y-axis to million €.
R1P23 Good point, the NPV concept does involve all revenues from all years, if we consider the entire forest area one enterprise. I’ve referred to it as “The development of NPV over time (i.e. the NPV in each year is the sum of all discounted costs and revenues in the preceding years and the current year, discounted to the start year 2016)” in the text and “NPV development over time in the 100-year planning horizon by year” in Figure 5.
Table 13: what are the area and volume numbers, sums for the whole 100 year period?
R1P24 Yes, these are the total area and total volume for the entire period, this has ben changed in table 13 and table 14.
P18L390 Difference compared to what, between scenarios?
R1P25 Yes, difference between scenarios. Clarified in text.
P18L39: do you mean difference between scenarios?
R1P26 Which line does this refer to? If same as above, then yes.
>>If different line – then no change<<
P18L392ff: Why is that? You did not use dynamic prices in BAU, did you, so this seems counter-intuitive and needs an explanation.
R1P27 The BAU scenario is a control scenario without dynamic prices and climate change P11 L278
But yes, the text could explain in more depth that CC negatively affects commercially valuable species more than it benefits marginally valuable lodgepole pine.
I’ve added sentence to explain it:
The CC factors reduced growth for commercially valuable conifers (i.e. spruces, firs and Doulas fir). Thus, the higher the CC effect, the less volume could be harvested from these species, reducing NPV
P18L405: I may have missed it, but I didn’t see you mention certification issues specifically in the methods. Should be mentioned there.
R1P28 Well it’s not really an issue for certification as the rules are rather fuzzy on the topic e.g. “take every step necessary to limit the use of fertiliser”. Coillte has double certification (FSC and PEFC) and they fertiliser a lot of area and maintain their certification.
I’ve clarified the sentence
Table 14: consider showing NPV as 1000 Eur to increase readability. Also harvest volume could be shown in 1000 m3. Why is the relative NPV for S3_noDP 1.09 when the absolute value is in fact smaller than BAU?
R1P29 Fair point, I’ve changed NPV and harvest volume to 1000s in table 14 and 13.
I’m not sure why the relative NPV for S3_noDP value was calculated wrong – my Excel sheet has the correct value. I’ve changed this value and checked all values in table 13 and 14 to ensure they are correct.
P18L406: I don’t think you should bring up new results in the discussion section. If you want to make this point, you need to mention the results of these extra runs already in the methods & results.
R1P30 Good point, allowing fertilisation is a “what-if” scenarios that was not really the scope of the study. The comment has been removed.
P19L414: This information needs to be mentioned already in the methods, I think.
R1P31 About bufferzones? I’ve included a mention of it in Table 5, under bufferzones.
P19L422: harvesting and reforestation, even with same tree species, can be considered to be quite a big change in forest structure. I guess you mean tree species distribution?
R1P32 I agree, the word structure has been changed to composition throughout the paper where referring to “changes in forest structure”. But as with the comment above (P15L347ff) I think composition might be a more appropriate word since it refers to things as bufferzones and native woodland sites.
>>Manuscript changes implemented under Reviewer 2 comment R2P12<<
P19L424: ‘normal age-class distribution’ – do you mean even, i.e. that your input data had an uneven age-class distribution?
R1P33 Thanks! Uneven is the right word, this has been changed.
P19L431: is your CSA representative of whole Ireland so that you can make this statement confidently?
R1P34 The CSA is representative for Western Ireland. I’ve rephrased the sentence and added a citation to Cabrera-Berned & Nieuwenhuis, as their study indicate revenue loss for most of the country.
P19L448ff: for which scenario and which time point are these prices?
R1P35 These values are using current prices. I’ve changed the wording to indicate that they are present-day prices.
P19L457-459: which study do you mean here, the Keenan et al study or yours?
R1P36 This is the Cabrera-Berned & Nieuwenhuis study. Sentences rewritten and citation added to make it clearer.
P20L460: increase compared to what?
R1P37 Compared to Climadapt’s default soil map. This has been specified in the text.
P21L539: Reduction compared to what?
R1P38 Compared to current growing conditions, this has been added.
Reviewer 2 Report
This research paper analyzes the relative effect that changes in climate and timber prices are likely to have on timber value (NPV) in an Irish peatland forest under alternative EU energy policy scenarios (GLOBIOM). The Introduction is clear and fully referenced. Methods are sound and make use of the best available knowledge. Results are clear but their presentation must be improved (see the attached pdf). The Discussion evaluates the results in the light of the available literature for Ireland and comparisons are attempted for other European contests. However, the authors must pay attention not to discuss results that they do not expose in the Results´ section. Overall, I believe that this paper represent a good case study to evaluate the effects of global change on forest ecosystems and specifically of the scenarios designed for climate change adaptation through implementation of bioenergy national and international regulations. Therefore, I recommend its publication after accomplishing the minor changes and answering the questions exposed in the attached pdf.

Author Response
Response to Reviewer 2 comments (Round 1)
Authors note: Answers and the corresponding manuscript edits (i.e. tracked changes) in to Reviewer 2 comments are marked R2 P X (Reviewer 2 Point X) in the revised manuscript. The below answers will have a corresponding comment in the revised manuscript and the cover letter detailing all changes made.
Point 1 L45: what was the reference status of forest cover?
R2P1 The current definition of forest in Ireland is: Forest is defined as land with a minimum area of 0.1 ha under stands of trees 5 m or higher, having a minimum width of 20 m and a canopy cover of 20% or more within the forest boundary; or trees able to reach these thresholds in situ.
However, the source does not state if this was the definition used in 1908.
>>No change in manuscript<<
Point 2 L53: The relationship between YC and mean annual volume increments must be specified in the paper.
R2P2 Added a footnote to define YC the relationship with MAI, I thought it’d be too much text for a parenthesis. Text in footnote.
[1] YC is the maximum mean annual increment of cumulative timber volume production for a species on a site, given in m3 ha-1 yr-1. Mean annual increment starts at zero and increases as the forest stand grows older, after it reaches maximum mean annual increment, i.e. the YC, mean annual increment declines. Irish plantation conifers reach maximum mean annual increment between 50 and 60 years of age, depending on the species.
Point 3 L60: by 2100?
R2P3 Yes, this has been added.
Point 4 138: do you mean, analyse the impact of global change scenarios? Rephrase the question.
R2P4 Fixed. Now the sentence is: analyse the impact that global change scenarios (representing different levels of CC and mitigation efforts) will have on forest management approaches and forest landscape structure. analyse the impact that global change scenarios (representing different levels of CC and mitigation efforts) will have on forest management approaches and forest landscape structure.
Point 5 L146: an excessive amount of achronyms make more difficult for the reader to understand the text. This is an online journal, so you can be more explicit.
R2P5
Removed acronyms: RSPS, SCA, SPA, NHA, pNHA, BFC, SS
The acronyms IIASA, GLOBIOM, RCP, IPCC are only used in one section of materials and methods, and I believe these acronyms are commonly used, rather than the full name.
Used throughout the paper: CSA, YC, NPV, FPM, CC, DP, SFM
Let me know if this is ok!
Point 6 L160: As you expect a change in climate you must refer to a thirty year reference period here for these measurements.
R2P6 This reference period is 1981-2010, which has been added.
Point 7 in Table 1: Do you mean public forest?
R2P7 Changed to Public (Coillte)
Point 8 L185: Specify what exactly is a Woodstock model in the legend.
R2P8 Clarified by calling it “Woodstock forest management model” since Woodstock could be utilised in other sectors beside forestry.
Point 9 L201: specify which shp you are talking about here.
R2P9 Clarified in text:
The ESRI shapefile containing spatial location of forest stands and the linked database with site information and forest inventory data was produced using ArcGIS 10.4
Point 10 L224: yield capacity must be related to growth units (volume, biomass) to hepl the reader to understand the text. YC can be used only for comparison among species.
R2P10 It doesn’t refer to the maximum capacity of a species to grow, but rather what growth and yield tables are available in Ireland. I’ve clarified this in the table caption.
Point 11 L302: replace with the names of the scenarios
R2P11 Added full scenario names
Point 12 L347: this is actually a change in forest composition. Chnages in forest structure imply changes in age and diameter classes.
R2P12 Changed to “forest composition”. Reviewer suggested changes on L348-349 approved.
Point 13 L 350: Isn´t instead this change determined by the dofferent scenarios?
R2P13 On a detailed level the change depends on the global scenarios i.e. the final area lodgepole pine. But overall all scenarios were affected by a similar, large change in forest composition.
I renamed the heading of 3.1 to “Change in forest composition in the global scenarios”, rather than “Change in forest composition from global scenarios” – I thought this heading would better reflect that a very similar change in forest composition took place regardless of potential future.
Point 14 L357/Figure 4: are they represented in the figures?
R2P14 Native woodland site is the small sliver between “Total buffer” (blue) and “Sitka spruce mixtures” (light-green). Native woodland site accounts for about 0.7% of the total area so it is does not stand out.
>>No change marked in manuscript. Although figure was made clearer<<
Point 15 L377: What is this?
R2P15 A duplicate cross-reference must have been introduced somehow, it has been deleted.
Point 16 L379: cumulative NPV
R2P16 Reviewer 1 commented that the wording “cumulative NPV” was strange. NPV does involve all revenues from all years, if we consider the entire forest area one enterprise. This has been changed to “total NPV”
>>Not changed marked in document, Reviewer 1 suggested word change<<
L390: USE DP for Dynamic Prices
R2 DP Changed throughout manuscript.
Point 17 L: the values for these three base scenarios must be reported together with the noCC and noDP scenarios to allow comprisons.
R2P17 Base scenario values added to the table. Table caption changed to match previous table (Cumualtive NPV, total clearfell area, total extracted harvest volume.
Point 18 L406: How is it possible to discuss missing results? Add these results or remove these considerations.
R2P18 Good point, allowing fertilisation is a “what-if” scenarios that was not really the scope of the study. The comment has been removed.
Point 19 L 490: For what concerns forest restoration through rewetting, you can cite:
Mazziotta, A., Heilmann-Clausen, J., Bruun, H. H., Fritz, Ö., Aude, E., & Tøttrup, A. P. (2016). Restoring hydrology and old-growth structures in a former production forest: modelling the long-term effects on biodiversity. Forest ecology and management, 381, 125-133.
R2P19 Thank you! I’ve added reference for the findings that forest restoration increase biodiversity. Successful rewetting of blanket bogs reduces species richness, but it provides habitat for a small number of rare species.
L499: change performs to perform
R2 perform Done
Point 20 L511: But this has also to do with wood value, right?
R2P20 Yes, comment on that added after the comparison of Sitka spruce and lodgepole YC change.
Text changed from:
Productivity reduction does not necessarily mean a species should be changed. If Sitka spruce YC 14 suffers a 15% growth reduction it will still grow better than lodgepole pine YC 10 on the same site that benefits from a 10% growth increase, as Sitka spruce would have a YC of 11.9 and lodgepole pine a YC of 11.0.
To:
Productivity reduction alone does not necessarily mean a species should be changed. If Sitka spruce YC 14 suffers a 15% growth reduction it will still grow better than lodgepole pine YC 10 on the same site that benefits from a 10% growth increase, as Sitka spruce would have a YC of 11.9 and lodgepole pine a YC of 11.0. However, depending on the DP it might be favourable to plant one species over the other.

Round 2
Reviewer 1 Report
I only managed to have a quick look, but it seems that the manuscript has been improved significantly. Below are some minor remarks:
P8L223 ff: Table caption missing, and bracket after table – needs to be fixed.
P10 L250: Error with reference to table.
P14 Fig3: Figure captions are typically below the figure.
Author Response
Forests manuscript ID 447758 Round 2
Replies to reviewer 1
P8L223 ff: Table caption missing, and bracket after table – needs to be fixed.
R1P1 I've removed the bracket around the explanatory text and have brackets closer to the Table 3 reference. The table reference has been changed to reference the table number.
P10 L250: Error with reference to table.
R1P2 Table reference fixed.
P14 Fig3: Figure captions are typically below the figure.
R1P3 Formatting changed to caption is below text.
